# Plant Secondary Metabolite Biosynthesis and Transcriptional Regulation in Response to Biotic and Abiotic Stress Conditions

**Rahmatullah Jan** [1,2,†] , **Sajjad Asaf** [3,†] , **Muhammad Numan** [4] , **Lubna** [5] and **Kyung-Min Kim** [1,2,*]

1  Division of Plant Biosciences, School of Applied Biosciences, College of Agriculture & Life Science, Kyungpook National University, Daegu 41566, Korea; rehmatbot@yahoo.com
2  Costal Agriculture Research Institute, Kyungpook National University, Daegu 41566, Korea
3  Natural and Medical Science Research Center, University of Nizwa, Nizwa 616, Oman; sajadasif2000@gmail.com
4  Laboratory of Molecular Biology and Biotechnology, Department of Biology, University of North Carolina at Greensboro, Greensboro, NC 27412, USA; mnuman@bs.qau.edu.pk
5  Department of Botany, Garden Campus, Abdul Wali Khan University, Mardan 23200, Pakistan; lubnabilal68@gmail.com
*  Correspondence: kkm@knu.ac.kr
†  These authors contribute equally to this manuscript.

**Abstract:** Plant secondary metabolites (SMs) play important roles in plant survival and in creating ecological connections between other species. In addition to providing a variety of valuable natural products, secondary metabolites help protect plants against pathogenic attacks and environmental stresses. Given their sessile nature, plants must protect themselves from such situations through accumulation of these bioactive compounds. Indeed, secondary metabolites act as herbivore deterrents, barriers against pathogen invasion, and mitigators of oxidative stress. The accumulation of SMs are highly dependent on environmental factors such as light, temperature, soil water, soil fertility, and salinity. For most plants, a change in an individual environmental factor can alter the content of secondary metabolites even if other factors remain constant. In this review, we focus on how individual environmental factors affect the accumulation of secondary metabolites in plants during both biotic and abiotic stress conditions. Furthermore, we discuss the application of abiotic and biotic elicitors in culture systems as well as their stimulating effects on the accumulation of secondary metabolites. Specifically, we discuss the shikimate pathway and the aromatic amino acids produced in this pathway, which are the precursors of a range of secondary metabolites including terpenoids, alkaloids, and sulfur- and nitrogen-containing compounds. We also detail how the biosynthesis of important metabolites is altered by several genes related to secondary metabolite biosynthesis pathways. Genes responsible for secondary metabolite biosynthesis in various plant species during stress conditions are regulated by transcription factors such as WRKY, MYB, AP2/ERF, bZIP, bHLH, and NAC, which are also discussed here.

**Keywords:** alkaloids; bioactive; deterrent; herbivores; salinity; terpenoids

## 1. Introduction

Compounds produced by plants are categorized into primary and secondary metabolites (SMs). Primary metabolites, such as carbohydrates, lipids, and proteins, are directly involved in plant development and growth. In contrast, SMs are multifunctional metabolites that are typically involved in plant defense and environmental communication [1]. Furthermore, they are associated with plant color, taste, and scent. Critically, they are also involved in the responses of plants to stress. For example, SMs are involved in the termination of infection, whether biotic- or abiotic-related [2]. Along with their importance in biotic stress tolerance, plant SMs are also involved in mitigating abiotic stresses such as temperature, drought, salinity, and UV light stresses [3]. When faced with certain biotic

and abiotic stresses, plants can reduce morphological traits such as the number of leaves or branches, leaf area, height, and root volume [4]. Indeed, plants have a diverse array of defense mechanisms that allow them to cope with stress conditions, mitigate abiotic stress at the metabolomic level, and enhance SM accumulation during stress.Threat signals are recognized by plants' receptors and sensors, which enable defensive responses in order to protect them from these stresses. The accumulation of secondary metabolites is one of the responses (Figure 1).Transcriptional factors (TFs) play a role in plant defense control by detecting stress signals and directing downstream defense gene expression. Similarly, the plant's survival, durability, and productivity are all dependent on increased synthesis of secondary metabolites, known as elicitation. Various biotic (fungi, bacteria, etc.) and abiotic (exogenous hormones) elicitors are used to enhance secondary metabolites production in plants to protect them from stress stimuli (Figure 1).

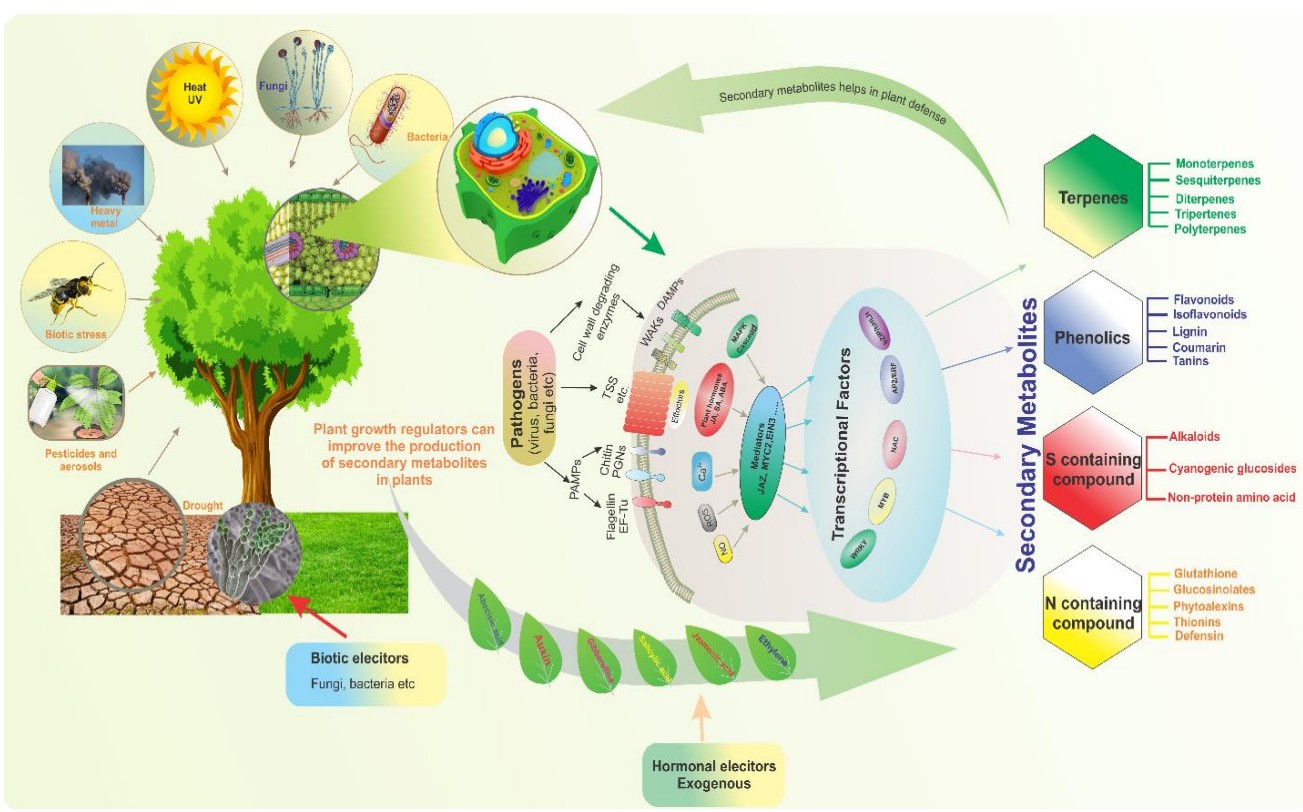

**Figure 1.** Diverse biotic and abiotic stresses affect plant growth and development; plants adopt various strategies and defense mechanisms to mitigate these stresses. To illustrate the modulation of secondary metabolism by various transcription factors, which are regulated by complicated upstream signaling pathways in response to stress, four plant secondary metabolite types involved in various modes of resistance are exemplified.

Plants produce SMs through several metabolic pathways that effectively respond to stress conditions. These pathways are initiated from primary metabolite pathways, which produce the ultimate precursors of SMs. The shikimate pathway is the initial pathway for biosynthesis of aromatic amino acids; it is activated in stress conditions to produced trypto-phan, tyrosine, and phenylalanine, which further enhance SM biosynthesis [5]. Different SMs accumulate conditionally in various plant parts depending on the stress condition. For example, phytoalexins have antimicrobial activities against phytopathogens and accumulate at high levels in leaves [6]. In addition to their antimicrobial properties, some SMs participate in the construction of polymeric barriers to pathogen penetration. In this regard, plants have a sophisticated recognition and signaling system that enables rapid recognition of pathogen attack and can initiate a dynamic defensive response. Accumulation of

SMs in response to stress conditions is regulated at the molecular level by various genes and transcription factors (TFs) including those of the phytohormonal pathway (Figure 1). Among all TFs, only a few modulate stress responses by mediating SM accumulation. This review is based on current knowledge of TF and gene-mediation of SM accumulation in response to stress conditions).

## 2. Biosynthetic Pathway of SMs

The precursors of metabolites are essentially produced in the Krebs cycle and shikimate pathway. The proposed SM biosynthesis pathway is shown in Figure 2. Primary metabolites are the critical precursors of SMs. Primary and SMs can be distinguished based on their chemical structure, function, and distribution in plants. The fundamental biosynthetic pathways of metabolites are conserved in the majority of plants, with most primary metabolites found in every tissue type. The maintenance of this metabolic core has led to the occurrence of a limited number of fundamental metabolic frameworks. Frequent glycosylation, methylation, hydroxylation, acylation, oxidation, phosphorylation, and prenylation, as well as fewer chemical alterations due to tailoring of enzymes, causes a wide range of modifications in basic structures. Based on biosynthesis pathways, SMs can be divided into three main groups: phenolic compounds synthesized in the shikimate pathway, terpenes synthesized in the mevalonic pathway, and nitrogen-containing compounds synthesized in the tricarboxylic acid cycle pathway [7].

**Figure 2.** Schematic of the secondary metabolite biosynthesis pathway.

Shikimic acid, the precursor of the shikimate pathway, is produced from a combination of phosphoenolpyruvate (from the glycolytic pathway) and erythrose 4-phosphate (from the pentose phosphate pathway). Phenylalanine, tyrosine, and tryptophan are produced in the shikimate pathway; these are the building blocks of protein synthesis and common precursors for plant SMs such as phenolics and nitrogen-containing compounds [6]. Phenylalanine is the common precursor of flavonoids, lignans, lignins, condensed tannins, and

phenylpropanoid/benzenoid volatiles; tyrosine further produces isoquinoline alkaloids, pigment betalains, and quinones (e.g., tocochromanols and plastoquinone); and tryptophan is the precursor of alkaloids, phytoalexins, indole glucosinolates, and the plant hormone auxin [8].

Seven steps are involved in the shikimate pathway to generate the end-product chorismate. In the first step, phosphoenolpyruvate and erythrose-4-phosphate are condensed to 3-deoxy-o-arabino-heptulosonate 7-phosphate (DAHP), which can formally be considered a 2-deoxy-D-glucose-6-phosphate derivative. In the second step, DAHP is exchanged with 3-dehydroquinate synthase to form the highly substituted cyclohexane derivative 3-dehydroquinate. The remaining steps in the shikimate pathway serve to add side chains and two of the three double bonds that convert this cyclohexane into a benzene ring (the hallmark of aromatic compounds). The third and fourth reaction in the shikimate pathway includes the dehydration of dehydroquinate to 3-dehydroshikimate, which is catalyzed by 3-dehydroquinate dehydratase (DHD), and the reversible reduction of 3-dehydroshikimate into shikimate using NADPH, which is catalyzed by shikimate dehydrogenase (SDH). DHD and SDH not only catalyze these respective reactions but are also part of the AROM complex in fungi. Furthermore, they are monofunctional in *Escherichia coli* but bifunctional in plants in the fused form of the DHD–SDH enzyme. The fifth step of the shikimate pathway is the formation of shikimate 3-phosphate. In this step, shikimate kinase catalyzes the phosphorylation of shikimate at the C3 hydroxyl group using ATP as a substrate. The sixth and seventh steps are catalyzed by 5-endolpyruvylshikimate 3-phosphate synthase (EPSP) and chorismate synthase, respectively. EPSP (also known as 3-phodphoshikimate 1-carboxyvinyltransferase) regulates the next to last step of the shikimate pathway by transferring the enopyruvyl moiety of phosphoenolpyruvate into shikimate 3-phosphate. Chorismate synthase is the final enzyme to participate in the shikimate pathway; it catalyzes the conversion of EPSP into chorismate, which is the precursor of SMs. In this final step, 1,4-anti-elimination of the 3-phosphate and C6-pro-R hydrogen from EPSP introduces the second double bond in the ring to produce chorismate. In higher plants, chorismate is the precursor of tryptophan, tyrosine, phenylalanine, salicylate, phylloquinone, and folate; it is regulated by enzymes such as chorismate mutase, iso-chorismate synthase, anthranilate synthase, and aminodeoxychorismate synthase [8].

## 3. Overview of SMs

### 3.1. Phenolic Compounds

Phenolic compounds, a group of SMs that are ubiquitous in plants and plant-derived foods, are essential for plant defense against parasites and pests [9]. They are characterized by their structure, which includes a minimum of one phenol ring. They are highly structurally diverse, containing simple molecules (e.g., vanillin, gallic acid, and caffeic acid) and polyphenols (e.g., stilbenes, flavonoids, and polymers). Phenolic compounds are usually present in plants in soluble or bound forms but they can also be categorized into subgroups according to their chemical structures. Soluble phenolic compounds are commonly synthesized in the endoplasmic reticulum and preserved in vacuoles, whereas bound phenolic compounds are produced by the transformation of soluble phenolic compounds to the cell wall, where they conjugate with the molecules of the cell wall through glycosidic and ester bonds [10]. General structure for each group of phenolic compound is presented in supplementary Figure S1.

### 3.1.1. Coumarin

Coumarins (2H-benzopyran-2-one), a class of simple phenolic compounds consisting of a large phenolic substance produced from the fusion of a benzene ring and $\alpha$-pyrone rings, are widespread in vascular plants. They were initially reported in the tonka bean (*Dipteryx odorata*) and subsequently reported in around 150 different species from 30 families [11]. About 1300 coumarins have been identified in plants as SMs; these function in a variety of capacities in many plants, typically in plant defense [12]. Along with defensive

effects on herbivores and fungi, coumarins have antimicrobial activity due to a specific bioactive group of molecules. They are distributed in all parts of plants; however, they are found at the highest levels in seeds [13]. Coumarins are naturally categorized into six types based on their chemical structure: simple (e.g., fraxin), furano (e.g., imperatorin), dihydrofurano (e.g., anthogenol), linear-type (e.g., grandivittin), phenyl (e.g., inophyllum), and bicoumarins (e.g., dicoumarol) [11].

### 3.1.2. Lignin

After cellulose, lignin is the most abundant biopolymer on earth and contributes to about 30% of the organic carbon in the biosphere. Lignin is one of the complex racemic heteropolymer components of the cell wall; it has a high molecular weight and is produced by the oxidative combinatorial coupling of primarily three p-hydroxycinnamoyl alcohol monomers (the so-called monolignols) [14]. Lignin is an important SM and broadly contributes to plant growth, development of tissues and organs, loading resistance, and biotic and abiotic stress tolerance. Lignification is a protective strategy of plant against pathogen and herbivores as it causes physical toughness to plant tissues which result the plant tissue non-digestible to insects and other herbivores.

### 3.1.3. Furanocoumarins

Furanocoumarin compounds are known for their phytotoxicity and are primarily found in species from Apiaceae and Rutaceae. Partly, furanocoumarins are synthesized in the phenylpropanoid and mevalonate pathways via the coupling of dimethylallyl pyrophosphate and 7-hydroxycoumarin. Usually, these compounds become toxic when they are activated by UV-A radiation, which leads to the activation of a higher electron energy state. Because of this activation, furanocoumarins enter the double helix of DNA and bind to pyrimidines; this causes cell death due transcription blockage and also possess mutagenic and carcinogenic properties [15]. Furanocoumarins also inhabit feeding behavior and physiological development of various insects [16]. It is found mostly in plant roots leaves and fruits and store in the form of essential oil. It has the great ability to sensitize cells to sunlight, and visible and ultraviolet light [17].

### 3.1.4. Flavonoids

Flavonoids, a diverse group of low-molecular-weight phenolic compounds, are found in the form of phytonutrients in the human diet and are ubiquitous in the plant kingdom (to date, ~8000 flavonoid molecules have been reported). They comprise one of the most characteristic classes of SMs in higher plants. The chemical structure of flavonoids usually consists of a 15-carbon skeleton with a phenyl ring and heterocyclic ring. Flavonoids have vital functions in plant defense and pigmentation systems. They also have wide range of health-promoting activities and a crucial ingredient of pharmaceutical, medicinal, and cosmetic products. For example, they can alter key cellular enzymatic functions because of their anti-inflammatory, antioxidative, anticarcinogenic, and antimutagenic properties, and they are efficient inhibitors of numerous enzymes [18]. Given the importance of flavonoids to plant life, they are classified in Figure 3.

### 3.1.5. Isoflavonoids

Isoflavonoids, a large group of diphenolic compounds classically defined as dietary antioxidants, include isoflavones, isoflavans and pterocarpans. They consist of a phenyl ring combined with a heterocyclic C-ring and another B-ring fused at the C3 position (unlike flavonoids, in which the B-ring is placed at the C2 position) [19]. Isoflavonoids are present in a limited group of plants and mostly found in leguminous species. They are naringenin derivatives that are important for nitrogen-fixing nodule formation and therefore the promotion of symbiotic rhizobia [20]. During plant–microbe interactions, isoflavonoids may be involved in the development of phytoalexins [21]. Some isoflavonoids have been reported to be present in microbes.

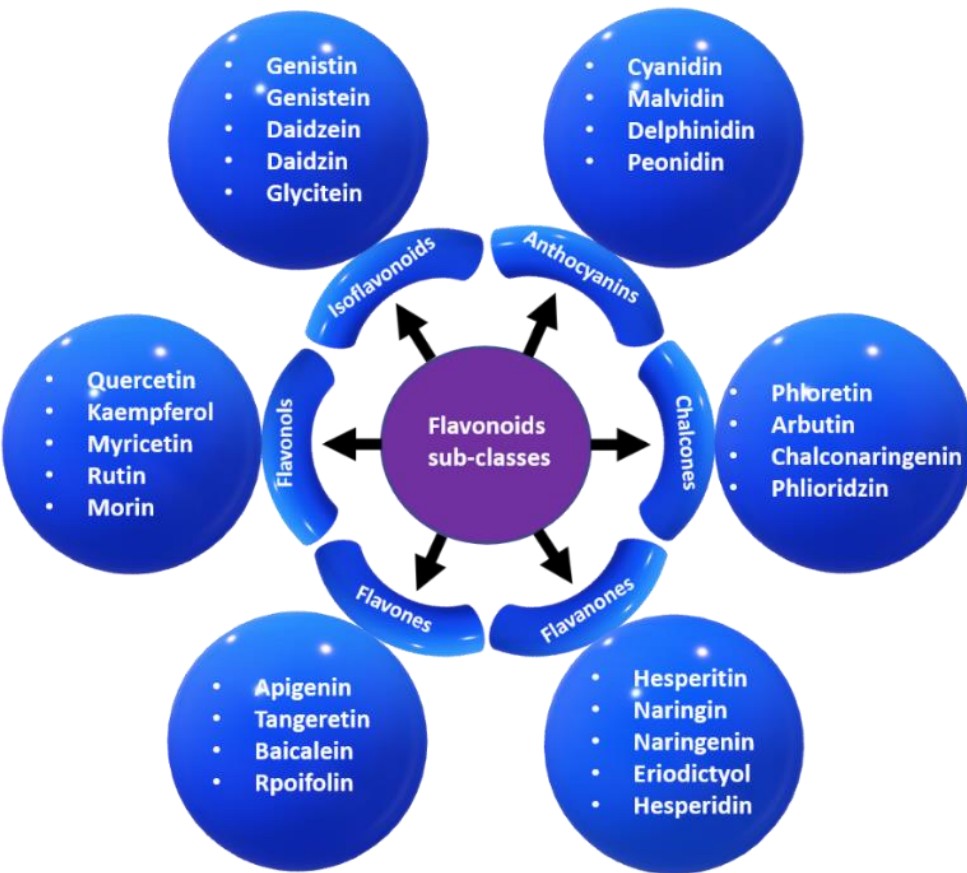

**Figure 3.** Classification of plant flavonoids.

### 3.1.6. Tannins

Tannins are the third essential group of phenolic compounds. They are rich in oligos and polymers and can make complexes with proteins, minerals, cellulose, and starch. Tannins can be divided into two subgroups, condensed and hydrolysable tannins, which have molecular masses of 600–3000. In general, tannins are toxins that have the potential to affect the growth and survival of most herbivores; therefore, they act as repellents to various grazing animals [15]. Tannins such as chlorogenic acid and protocatechuic acid also play major roles in the resistant of plants to various diseases. Environmental factors, such as excessive $CO_2$ in atmosphere, high temperature, light conditions, and nutrient variation, can significantly alter the accumulation of tannins in the plants [22].

### 3.2. Terpenes

Terpenes or terpenoids, the major class of SMs in plants, are formed from the derivatives of glycolytic or acetyl CoA intermediates. They are lipid-soluble and have vast structural diversity (more than 50,000 different structures) [23]. Terpenes are the essential building blocks of various complex phytohormones, sterols, and pigments, and they are primarily responsible for their aromas and physiological effects. Although some terpenes have a pleasant aroma and act as pollinator attractants, most act as defensive toxins and herbivore deterrents. See supplementary Figure S1 for general structure of each group of terpenes.

### 3.2.1. Monoterpenes

Monoterpenes, a major group of terpenes, are associated with the 10-carbon-containing isoprenoids and are extensively used in health care products and cuisine. Natural monoterpenes and their derivatives have critical pharmaceutical properties such as antioxidant,

anticancer, antibacterial, antifungal, antiviral, antiaggregating, anti-inflammatory, anti-histaminic antispasmodic, and local anesthetic activities [24]. Monoterpene derivatives (e.g., β-pinene, α-pinene, myrcene, and limonene) are toxic to insects; they accumulate in the resin ducts in twigs, needles, and trunks. They are also potentially involved in growth regulation and heat tolerance. In the environment, monoterpenes give a specific identity to their respective plants. Examples include citral, which is a monoterpene present in lemons that gives them their special smell, and thymol, which forms part of the flavor of mandarin oranges. Some monoterpenes are present in the scent of flowers and can attract plant pollinators, whereas others are repellents of predators [25].

### 3.2.2. Sesquiterpenes

Sesquiterpenes, another major class of terpenes, contain three units of isoprene and are found in cyclic and acyclic forms. They occur in oxygenated forms or in the form of hydrocarbons such as alcohols, lactones, aldehydes, and ketones. More than 5000 sesquiterpene compounds have been identified in the plant kingdom, most commonly in Asteraceae [26] like α-bisabolene and β-caryophyllen. Sesquiterpenes (like costunolides) are characterized by a five-membered lactone ring. They are known for their defensive action against herbivores and ability to repel the feeding of mammals and insects [27]. Sesquiterpenes also have a regulatory role in the initiation and preservation of bud and seed dormancy and can act as transcriptional activators [28].

### 3.2.3. Diterpenes

Diterpenes, which are composed of two terpene units, are biosynthesized by plants, animals, and fungi through the isoprenoid pathway (i.e., the mevalonate pathway) using geranylgeranyl pyrophosphate. Depending on their core skeleton, they can be divided into linear, bicyclic, tricyclic, tetracyclic, penta, and macrocyclic forms. Diterpenes are the precursors of biologically essential metabolites such as retinal, retinol, and phytol, which have anti-inflammatory and antimicrobial activities. They show significant biological activity in various medicinal plants; for example, ingenol-3-angelate and paclitaxel are effective against cancer. Other diterpenes, such as forskolin, salvinorin A, triptolide, carnosic acid, and ginkgolide B, are potentially involved in analgesic, cardioprotective, antioxidant, and anti-inflammatory activities [29]. Gibberellins (phytohormones) are important diterpenes that have a key function in seed germination, flower and fruit setting, leaf expansion, biomass production, stomatal conductance, and $CO_2$ fixation [15].

### 3.2.4. Triterpenes

Triterpenes, a group of terpenes consisting of six isoprene units, have been identified in more than 200 skeletons, with about 20,000 different terpenes reported to date [30]. Triterpenes contain 30 carbons and can be found in linear, dicyclic, tricyclic, tetracyclic, and pentacyclic forms that are mainly derived from squalene [31]. The prominent members of this group show antiviral, antibacterial, antifungal, antioxidant, anticancer, and anti-inflammatory activities [32]. They play important roles in pest and pathogen control, as well as food quality in crop plants, and are also contained in pharmaceutical, food, and cosmetics products [33].

### 3.2.5. Polyterpenes

Polyterpenes, such as tetraterpenes, have high molecular weights, as they comprise several hundred isoprene units. Rubber is a polyterpene polymer consisting of many repeating isoprene units. Usually, rubber is present in laticifers, in which it provides protection against herbivores and enables wound healing as a defense mechanism [34]. Polyterpenes act as viscosity diluents like waxes as well as act as a co-tackifiers [35].

### 3.3. Sulfur-Containing SMs

Sulfur-containing secondary metabolites are a relatively minor group of plant SMs including about 200 compounds. The group encompass the well-known glucosinolates and their breakdown products such as thiocyanates, isothiocyanates, epithionitriles, and oxazolidinethiones [36]. Sulfur-containing SMs are essential for protein synthesis: with a lack of sulfur, protein synthesis is reduced. The symptoms of sulfur deficiency are similar to those of nitrogen deficiency, but are only seen in the young leaves of plants [37]. Sulfur-containing SMs have a significant effect on plant health because plants cannot utilize nitrogen properly without having sufficient sulfur [38]. Studies have reported that sulfur- and nitrogen-containing SMs are affected by sulfur and nitrogen supply, i.e., the ideal amount of each respectively enhances the ability of the plant to deal with various environmental stresses. Sulfur-containing SMs include glutathione, glycosphingolipid, phytoalexins, allinin, thionins, and defensins, which are directly or indirectly linked to the plant defense system [15].

### 3.4. Nitrogen-Containing SMs

Many plant SMs contain nitrogen in their skeleton. Nitrogen-containing SMs are synthesized from common amino acids and can be divided into main four categories: alkaloids, cyanogenic glycosides, glucosinolates, and nonprotein amino acids. SMs occurrence in plant species and their biological activities are presented in Table 1.

**Table 1.** The occurrence and biological activities of secondary metabolites (SMs) in various plant species.

| SM Name | Class | Plant | Biological Activity | References |
|---|---|---|---|---|
| **Terpenes** | | | | |
| Iridoids | Monoterpenes | *Eucomis autumnalis/Scrophularia ningpoensis* | Drought stress | [39] |
| Cineole | Monoterpenes | *Mentha spicate/Thymus vulgaris* L. | Drought stress | [40] |
| Tanshinone | Diterpenes | *Salvia miltiorrhiza* | Hyperosmotic stress | [41] |
| Retinol | Diterpenes | *Daucus carota* | Oxidative stress | [42] |
| Carnosic acid | Diterpenes | *Salvia officinalis* L. | Oxidative stress/Drought stress | [43] |
| Phytol | Diterpenes | *Solanum lycopersicum* | Light stress/Temperature stress | [44] |
| Abietic acid | Diterpenes | Leguminous/Pine trees | Antipathogen | [45] |
| Gikolides | Diterpenes | *Ginko biloba* | Biotic stress | [46] |
| Menthol | Monoterpenes | *Mentha spicata* L./*Arabidopsis thaliana* | Antifungal | [47] |
| Catalpol | Monoterpenes | *Plantago lanceolata* | Drought stress | [48] |
| Pyrethroids | Monoterpenes | *Chrysanthemum morifolium* | Anti-insect | [49] |
| α-pinene/β-pinene | Monoterpenes | *Chrysanthemum morifolium*/Conifers | Antipest/Drought stress | [49] |
| Artemisinin | Sesquiterpene | *Artemisia annua* L. | Light stress/Cold stress/antimalaria | [50] |
| Gossypol | Sesquiterpene | *Gossypium* L. | Antipathogen/Antipest | [51] |
| Limonoids | Triterpenes | Citrus fruits | Antiherbivore | [49] |
| Ginsenosides | Triterpenes | *Panax quinquefolius* | Light stress | [52] |
| Sterols | Triterpenes | *Asclepias syriaca* | Antiherbivore/Anti-insect | [53] |
| **Phenolic Compounds** | | | | |
| Cyanidin/Peonidin/Malvidin/Pelargonidin/Delphinidin | Flavonoids | Cabbage/Radish/Lettuces | Abiotic stresses | [54] |
| Tangeretin/Nobiletin | Flavonoids | Citrus | Abiotic stresses | [55] |
| Leteolin/Chrysoeriol/Apigenin | Flavonoids | *Apium graveolens/Thymus vulgaris* | Biotic/Abiotic stresses | [55] |
| Quercetin/Qaempferol/Myrecetin | Flavonoids | *Allium cepa/Allium porrum/Brassica oleracea var.* | Oxidative stress/Antipathogen | [56] |

**Table 1.** *Cont.*

| SM Name | Class | Plant | Biological Activity | References |
|---|---|---|---|---|
| Genistein/Daidzein/Glycitein | Flavonoids | *Glycine max* | Oxidative stress | [57] |
| Podophyllin/Urushiol | Lignans | *Podophyllum* | Biotic/Abiotic stress | [58] |
| Pinoresinol | Lignins | All land plants | Cold stress | [59] |
| Oleuropein | Polyphenolic | *Olea europaea* | Salt stress | [60] |
| Gallic acid | Phenolic | *Pisum sativum* | Antifungal | [56] |
| Ferulicacid | Phenolic | *Oryza sativa* | Antifungal | [61] |
| Gallotannin | Tannin | *Quercus robur/Tsuga/Lotus corniculatus*/Legumes | Oxidative stress/Antibacterial | [62] |
| Tannic acid | Tannin | *Gossypium* | Salt stress/Oxidative stress | [63] |
| Umbelliferone/Esculetin/Scopoletin | Coumarins | *Olea europaea/Avena sativa Apiaceae/Asteraceae/Fabaceae* | Biotic/Abiotic stress | [64] |
| Psoralin | Furanocoumarins | *Psoralea corylifolia L* | Biotic stress | [65] |
| Bergaptene | Furanocoumarins | *Ammi majus* | Antibacterial | [66] |
| Gallic/Salicylic acids | Phenolic acids | Berries/Nuts | Antiviral | [67] |
| Juglone | Napthoquinones | *Juglans regia* | Chemical detoxificante/Abiotic stress/Oxidative stress | [68] |
| Plumbagin | Napthoquinones | *Plumbago* spp./*Drosophyllum* spp. | Antimicrobial/Anti-insect | [69] |
| Emodin | Anthraquinones | Higher plants | Antiherbivore/Antipathogen/Light stress | [70] |
| Caffeic | Hydroxycinnamic acids | *Prosopis farcta* | Heavy metal stress/Drought stress/Salt stress | [71] |
| Ferulic acids | Hydroxycinnamic acids | *Vitis vinifera* L./*Chrysanthemum morifolium* | Drought stress | [72] |
| Bergenin | Isocoumarins | *Saxifraga ligulata* | Antiviral/Oxidative stress | [73] |
| Eusiderin | Neolignans | *Eusideroxylon zwagery* | Antifungal | [74] |
| Mangiferin | Xanthones | *Mangifera indica* | Oxidative stress/Biotic/Abiotic stress | [75] |
| Amentoflavone | Biflavonoids | *Selaginellaceae/Cupressaceae/Euphorbiaceae* | Oxidative stress/Antiviral/Antifungal | [76] |
| Agathisflavone | Biflavonoids | *Anacardium occidentale* | Oxidative stress/Antimicrobial/Antiviral | [77] |
| **Sulfur-containing SMs** | | | | |
| Defensins/Thionins/Lectins | | Higher plants | Antifungal/Antibacterial | [78] |
| Brassinin | Phytoalexin | *Brassica campestris* L. | Biotic/Abiotic stress | [79] |
| Wasalexins | Phytoalexin | *Wasabia japonica* | Biotic/Abiotic stress | [80] |
| Camalexin | Phytoalexin | *Camelina sativa* | Biotic stress | [79] |
| Glucoraphanin | Glucosinolates | *Brassica oleracea* | UV-B stress | [81] |
| Neoglucobrassicin | Glucosinolates | *Brassica oleracea* | UV-B stress | [81] |
| Glucobrassicin | Glucosinolates | *Brassica rapa* | Wounding Stress | [82] |
| Sinigrin | Glucosinolates | *Brassica oleracea* | Salt stress | [83] |
| **Nitrogen-containing SMs** | | | | |
| Narkotine/Codeine/Morphine | Alkaloid | *Papaver somniferum* | Drought stress | [84] |
| Monocrotaline | Alkaloid | *Crotalaria* | Antiherbivore | [85] |
| Tomatidine | Alkaloid | *Solanum lycopersicum* | Salt stress | [86] |
| Senecionine | Alkaloid | *Senecio jacobaea* | Antiherbivore | [87] |
| Vincristine/Vinblastine | Alkaloid | *Catharanthus roseus* | UV-B stress | [88] |
| Pyrrolizidine | Alkaloid | *Panax quinquefolius* | Antiherbivore | [89] |
| Mimosine | Nonprotein amino acids | *Mimosa pudica* | Antiherbivore/Antimicrobial | [90] |

**Table 1.** *Cont.*

| SM Name | Class | Plant | Biological Activity | References |
|---|---|---|---|---|
| Citrulline | Nonprotein amino acids | *Cucumis melo* L. | Drought stress/Salt stress | [91] |
| Pipecolic acid | Nonprotein amino acids | *Calliandra* | Antiherbivores | [92] |
| Canavanine/Azetidine-2-carboxylic acid | Nonprotein amino acids | Legumes | Antiherbivore/Antipathogen | [15] |
| Amygdalin | Cyanogenic glucoside | Almonds/Apricot/Cherries/Peaches | Antiherbivore | [15] |
| Dhurrin | Cyanogenic glucoside | *Sorghum bicolor* L. | Drought stress/Antiherbivore | [93] |
| Linamarin/Lotaustralin | Cyanogenic glucoside | *Lotus japonicus* | Antiherbivore/Drought stress | [94] |

## 4. Responses of SMs to Abiotic Stress

SM accumulation, which is dependent on plant growth condition and influences related metabolic pathways, is often associated with stress conditions, and may include signal molecules and elicitors. Due to the sessile nature of plants, they are regularly exposed to a wide range of environmental stresses, such as temperature, drought, salinity, alkalinity, UV, pathogens, and herbivores, which can cause serious damage to the plants [95]. Exposure to abiotic stress causes various common reactions that can lead to cellular dehydration, which in turn results in the generation of osmotic pressure and removal of water from the cytoplasm to vacuoles. External factors can adversely affect some process associated with growth and development in plants, and their ability to biosynthesize SMs ultimately leads to variation in their overall phytochemical profiles, which play important roles in the production of bioactive substances [96,97]. In other words, plant SMs are regularly synthesized in response to environmental stress; hence, secondary metabolism, in part, determines the ability of plants to adapt and survive in response to environmental stimuli and stress during their lifetime, while also enabling ecological relationships to take place between plants and other organisms. Below, we briefly discuss the synthesis of SMs in association with the major abiotic stresses.

### 4.1. Temperature

Global temperature increased by ~0.47 °C in the twentieth century. Global warming, it is estimated, will substantially affect the production of plant SMs. This is because high, as well as low, temperatures are harmful abiotic stresses that can impact the survival of plants; therefore, adapted plants alter their metabolism when faced with such temperatures to increase the level of essential metabolites and thereby tolerate unfavorable conditions [98]. High temperatures enhance the biosynthesis of alkaloids; in contrast, alkaloid synthesis (e.g., morphinane, phthalisoquinoline, and benzylisoquinoline) is inhibited at low temperature in plants such as *Papaver somniferum*. For example, the alkaloid hydroxycamptothecin is involved in heat-shock tolerance; hydroxycamptothecin accumulation was reported to increase 6-fold in the leaves of *Camptotheca acuminata* seedlings during incubation at 40 °C for 2 h [99]. Various cultivars of *Lupinus angustifolius* exposed to high temperature in the field also significantly induce alkaloid accumulation [100]. Conversely, *Catharanthus roseus* leaves show reduced levels of vindoline and catharanthine when exposed to low temperatures [101]. Temperature is also correlated with terpenoids in coniferous and deciduous plants: emission of isoprenes from *Quercus rubra* and *Quercus alba* at high temperatures was double that recorded at low temperatures [102]. In response to high temperature, terpenes accumulate at high concentrations in *Daucus carota* with the exception of α-terpinolene, which decreases with increasing temperature. In the leaves of *Zea mays* seedlings, anthocyanin accumulation is enhanced with the duration and severity of low temperature. In *Rhodiola rosea*, temperature alters SM synthesis and induces SM accumulation in combination with heavy metal stress [103]. Phenylamides have reactive oxygen species (ROS) scavenging properties during stress conditions; they are produced in tobacco and beans when the plants are exposed to heat shock and water stress [104]. Low temperature en-

hances the synthesis of phenolic compounds and their subsequent integration in the plant cell wall as lignin or suberin, while plants adapted to cold environments are associated with the synthesis of chlorogenic acid at high levels [59]. Similarly, temperature influences the synthesis of ginsenosides in the root hairs of *Panax ginseng*, while *Melastoma malabathricum* synthesizes high levels of anthocyanin at low temperatures [105].

### 4.2. Salinity

Salinity is a major abiotic stress that leads to cellular dehydration; this causes ionic or osmotic pressure that increases or decreases the accumulation of specific SMs in plants. Salt stress may act as an elicitor of SMs for the protection of cells from the oxidative damage caused by ion accumulation at the cellular and subcellular levels; thus, SMs reduce the toxic effect of salinity [106]. Plant genotypic flexibility in diverse saline environments, both in vitro and in vivo, enables plants to synthesize SMs that are crucial for their survival in such conditions. Indeed, salinity has physiological, biochemical, morphological, and biosynthesis effects on plant SM profiles due to the induction of stress and defense response pathways involved in ROS generation and responsible for the modulation of SMs [107]. When salt stress occurs, the ability of roots to absorb sufficient water reduces, which may in turn lead to water scarcity and osmotic pressure caused by high accumulation of salt [108]. As a result of osmotic pressure, membrane functional stability, nutrient balance, and redox homeostasis are disturbed, which in turn affects the primary metabolites that are the precursors of SMs [109]. In recent decades, the physiological and molecular effects of salt stress on the production of important SMs in crop species have been reported. These SMs include terpenoids, flavonoids, alkaloids, steroids, and phenolics which function in plant defensive responses to salt stress and protect plant growth physiology [110]. For instance, anthocyanins were reported in higher concentrations during salt stress, while anthocyanin levels were reduced during salt stress in salt-susceptible species [111]. Salt-tolerant alfalfa increase their proline levels under salt stress, and a similar trend was found in *Lycopersicon esculentum* and *Aegiceras corniculatum*. Although polyphenols are generally associated with abiotic stress, polyphenol accumulation also increases in plant tissues under salt stress in various plants [112]. Furthermore, phenolic compounds increase in pepper plants under salt stress, which is consistent with the response of polyamines that are involved in the salt-stress response [113]. In addition, *Sesuvium portulacastrum* under salt stress reduces certain physiological process due to the synthesis of SMs: the strong antioxidant properties of these SMs play an important role in the plant's survival under saline stress [114]. Halophytes, which grows in saline soil and spend an extensive part of their life under salt stress, have high osmotic pressure that enables them to uptake sufficient water and synthesize SMs via various mechanisms [109].

### 4.3. Water (Drought and Flooding)

Drought, a prevalent and multidimensional abiotic stress in the plant kingdom, occurs widely in arid and semiarid areas throughout the world. Drought stress usually leads to morphological, physiological, biochemical, ecological, and molecular changes in plants; hence, it can adversely affect the quality and quantity of plant biomass [115]. During their evolution, plants have adapted to drought conditions by accumulating SMs, such as terpenes, alkaloids, and phenolic complexes, via ionic or osmotic stress induction; however, increased SM accumulation is usually accompanied by reduced biomass [116]. For instance, the concentrations of phenolic compounds in *Hypericum brasilience* and *Pisum sativum* increase when they are exposed to drought stress [117]. Phenolic compounds typically accumulate during drought stress because of changes in the phenylpropanoid pathway: most of the key genes of this pathway are stimulated by drought stress. For example, drought stress induces activation of the *PAL* gene in lettuce plants and expression of several genes related to flavonoid biosynthesis in *Scutellaria baicalensis* [118]. Similarly, accumulation of terpenes in *Salvia officinalis* increases during drought stress while accumulation of biomass is reduced [119]. Drought stress promotes oxidative stress, which in turn

enhances flavonoid biosynthesis; flavonoids act as antioxidants and protect plants from the effects of water scarcity. For example, kaempferol and quercetin levels increase in tomato plants under drought stress; these compounds can potentially detoxify the $H_2O_2$ molecules produced during drought stress. Some SMs are reduced, however, during water scarcity, e.g., saponins in *Chenopodium quinoa* [120]. Researchers can induce and test drought stress in correlation with SMS accumulation in vitro; culture medium (containing nutrients, carbon sources, and osmotic stabilizers) can be manipulated to produce drought conditions with associated effects on the metabolic processes that lead to SM accumulation [121]. For example, manipulation of culture medium significantly impacts accumulation of camptothecin in *Nothapodytes nimmoniana* and *Ophiorrhiza mungos*, phenolic compounds in *Bellis perennis*, and tropane alkaloids in *Brugmansia candida* [122,123]. The effects of drought on SMs in conifers have also been studied; total terpenes in the seedlings of *Pinus sylvestris* and *Pinus abies* were found to be about 32–39% and 35–45% higher in drought-affected plants than in control plants, respectively [124].

### 4.4. Radiation (Light, UV, and Ionization Radiation)

Light quality can also influence synthesis of bioactive compounds and SMs in plants. The basic components of light include photoperiod (length), power (sum), and quality (repeat/recurrence) [125,126]. Different plant species shows variations in their responses to quantity and intensity of light [127]. Because of the depletion of the ozone layer, concerns have arisen about the effects of UV radiation (UV-B: 280–320 nm), which has strong effects on the composition and concentration of plant SMs [127] including alkaloids [128], terpenoids, flavonoids [129], cyanogenic glycosides, tannins, and anthocyanins [130]. Plants are able to adapt to changes in light radiation by accumulating and releasing various SMs, such as phenolic compounds, triterpenoids, and flavonoids, many of which have high economic value and utilization due to their antioxidant properties [131,132]. Studies have shown that the duration of light radiation affects the regulation of levels of various phenolic phenylpropane derivatives in the *Xanthium* species. In comparison with a long period of light exposure, a short period of light exposure reduces caffeoylquinic acids by about 40% and approximately doubles the reduction in flavonoid aglycone content [133]. Similarly, in Pinus contorta, the concentration of anthocyanins was found to be lower when the plant was grown under short sunlight conditions compared with long sunlight conditions, whereas concentrations of proanthocyanins and flavan-3-ols concentrations were barely altered with variation in sunlight period [134]. In Ipomoea batatas, a significant increase in phenolic acids (e.g., hydroxybenzoic acids and hydroxycinnamic) and flavonoids (e.g., flavonols, anthocyanins, and catechins) content was observed after a long period of light exposure [135].

The synthesis of SMs is also affected by light quality. The phenolic compounds of *Lactuca sativa* are more sensitive to monochromatic light than combined light. In one study, increasing the proportion of red light affected antioxidant phenols, while phenolic compounds, including caffeic acid, chicoric acid, chlorogenic acid, and kaempferol, as well as ferulic acids, showed similar behaviors [136]. UV irradiation is a factor that in many cases enhances the synthesis of SMs; therefore, it can be used in cell and callus cultures [137–139]. In one study, production of vindoline and catharanthine from *Catharanthus* species was increased after UV-B radiation treatment [140]. Similarly, glycosyl flavonoid content was found to be significantly higher following UV irradiation treatment [137]. In addition, Regvar, et al. [141] reported the effect of UV irradiation on the concentration of quercetin catechin and rutin in *Fagopyrum tataricum* and *Fagopyrum esculentum*; they detected a significant increase in quercetin concentration in *F. esculentum* after UV irradiation treatment.

### 4.5. Gaseous Toxins

Air pollutants, such as sulfur dioxide ($SO_2$), can enter plant systems through the roots and stomata during photosynthesis and respiration. Different plants shows various

responses to SO$_2$, which include photosystem damage [142], variation in stomatal density, and changes in carbon fixation efficiency [143,144]. Owing to the importance of sulfur in many important pathways, stomatal uptake of SO$_2$ influences the metabolic profile of the plant. Glucosinolates, which are sulfur-containing metabolites, are thought to be important in sulfur storage, which provides a source of sulfur for the plant when sulfur is not present in the environment [145]. SO$_2$-induced fumigation has been shown to change anthocyanin synthesis, although the low abundance of flavon-3-ol transcripts after fumigation with SO$_2$ indicates that anthocyanin is not rapidly degraded [146]. When considering approaches to supplying sulfur to plants, there is an interest in understanding the effects of the sulfur-containing gas, hydrogen sulfide (H$_2$S). High dosages of H$_2$S can induce defoliation, leaf lesions, decreased growth rate, and tissue death in some plants [147]. H$_2$S is also a signaling molecule that promotes antioxidant activities in many plants in response to abiotic stresses. One study showed that application of H$_2$S changes the antioxidant potential of some plants [147]. In this study, incremental levels of sodium hydrosulfide (with H$_2$S as a donor) were applied to the Bronco cabbage (*Brassica oleracea*) to investigate potential physiological and antioxidative changes; lower treatment levels caused an increase in plant content of carotenoids, anthocyanins, flavonols, total phenolics, and sinigrin [147].

### 4.6. Heavy Metals

Heavy metals have become one of the main abiotic stress agents for plants given the influence of agrotechnology, industry, and their high bioaccumulation and toxicity [148]. Substantial data are available on the effects of heavy metals on plant growth and physiology; however, fewer data are available on the effects of heavy metals on SM production. Heavy metal-induced changes in the metabolic activity of plants can affect the production of sugars, photosynthetic pigments, and proteins. These effects occur because of the inhibition of enzymes involved in the production of these natural products [149]. Heavy metals can also change the synthesis of bioactive compounds by altering certain aspects of secondary metabolism [150]. For example, metals such as Fe, Ag, Ni, and Co have been reported to provoke the synthesis of SMs in various plants [151]. Similarly, in a specific study of *Brassica juncea*, the effective accumulation of metals (e.g., Fe, Cr, Zn, and Mn) significantly increased the oil content of the plant by up to 35% [152]. Following treatment with Cd$^{2+}$ and Cu$^{2+}$ in other studies, maximum accumulations of SMs such as shikonin [153] and digitalin [154] occur. In addition, the synthesis of betalains in *Beta vulgaris* also increases with Cu2+ treatment, while Cu$^{2+}$ and Co$^{2+}$ have stimulatory effects on the synthesis of SMs in this species [155]. Furthermore, Zn$^{2+}$ treatment (900 μM) enhances the production of lepidine in cultures of *Lepidium sativum*; however, Cu is more effective than Zn in increasing the product yield [156]. Similarly, in hairy root cultures of *B. candida*, cadmium chloride (CdCl$_2$) or silver nitrate (AgNO$_3$) provokes the overproduction of hyoscyamine and scopolamine [157]. Finally, in *Taxus* sp. cell cultures, the synthesis of taxol increases with lanthanum earth-metal treatment [158].

### 4.7. Pesticides and Aerosols

Despite the many benefits of pesticides, their use can affect nontarget organisms, including humans and plants, and thereby create abiotic stress conditions [159]. Although pesticide residues are controlled in plants to protect human health [160], this process does not reflect the fact that plant secondary metabolism can be controlled by pesticides. An important question therefore arises: can pesticides induce chemical changes in plants that affect human health [160]? Studies suggest that this might be possible at the time of application but is less critical at the time of consumption [160]. Typically, plant parts such as leaves, roots, and fruits are exposed to pesticides; when they enter the plant, however, pesticides reach different plant parts using the vascular system and cell-to-cell movement (depending on the pesticide's chemical and physical properties). Most plants have a specialized system for sequestration and degradation of pesticides [161]. The metabolic biotransformation of pesticides can be split into different phases. For example, the first phase includes epoxi-

dation, oxidation, hydroxylation, reduction, hydrolysis, O-dealkylation, N-dealkylation, desulfuration, dehydrogenation, and dehalogenation. Most of these chemical changes are mediated by enzymes whereas some arise from photochemical reactions. The second and third phase reactions involve conjugation of the pesticide and its metabolites with other natural constituents such as glutathione, malonic acid, and glucose [162,163].

To date, most pesticide research has focused on the effects of pesticides, their residues and metabolism, and their impact on human health and the environment. However, the effects of pesticides on plant SMs (both targeted and nontargeted) are largely unknown. Most published data relate to the effects of fungicides and herbicides on flavonoid content and some terpenoid metabolism in plants [164,165]. Pesticides may also disturb secondary metabolism indirectly by eliminating nontarget plants species that compete for nutrients and light or serve as habitats for herbivores and pathogens [166]. Pesticide-related changes in plant SMs, such as phenols, not only affect ground-surface organisms and ecological functions but also influence the rhizospheric community if pesticides are applied to plants in the field [167]. Furthermore, after a plant's death, pesticide-induced production of plant phenols may affect its availability as a food for soil organisms or slow the decomposition rate of soil organic matter because the phenols can be persevered for weeks to months after death [168].

## 5. Response of SMs to Biotic Stress

Different methods are used for the biological control of phytopathogens, but one of the most important is the use of medicinal plant extracts, which is aimed at eradicating crop diseases caused by bacteria, fungi, and viruses [169,170]. Thus, it is important to study the secondary metabolism of plants in this context, i.e., understanding which phytochemical substances are produced by plants and the biological activities of these phytochemicals [171].

### *5.1. Biotic Elicitors*

Elicitors can be characterized as stress factor substances that when added to a living system in small amounts induce or improve the biosynthesis of specific compounds associated with the adaption of plants to stressful conditions. Elicitors may be biotic or abiotic (Figure 1). Biotic elicitors are organic substances that typically contain carbohydrates and develop their signal effects at low concentrations [172]. Here, we discuss types of both biotic and abiotic elicitors.

#### 5.1.1. Polysaccharides

In medicinal plants, SM production can be triggered using biotic elicitors. In cell suspension cultures of *P. ginseng*, saponin content significantly increases when oligogalacturonic acid is used as an elicitor [173]. In addition, production of the natural naphthoquinone shikonin is induced by the polysaccharide agaropectin in the cells of *Lithospermum erythrorhizon* [174]. Similarly, plumbagin content in *Plumbago rosea* is increased via chitosan treatment [69]. Chitin or chitosan also induce the biosynthesis of fluoroquinolone coumarin alkaloids in shoot cultures of *Ruta graveolens* [175]. Furthermore, using chitosan in cell cultures of *Vitis vinifera* enhances the production of viniferins and trans-resveratrol [176]. Additionally, production of naphtodianthrone and phenylpropanoid is enhanced by chitin treatment in cell suspension cultures of *Hypericum perforatum* [177].

#### 5.1.2. Yeast

Yeast extract has been used for decades as a biotic elicitor for the production of SMs. In tomato plants, yeast induces the production of ethylene, while in *Phaseolus vulgaris* it induces bacterial resistance [178]. It also induces tanshinone when used as an elicitor in *Perovskia abrotanoides* root tissue culture [179].

### 5.1.3. Fungal Elicitors

Pathogen-induced biotic elicitors are mainly used to activate plant defense systems (Table S1). For the production of phenylpropanoids/flavonoids in plant cells, the use of fungal pathogenic and nonpathogenic elicitors is one of the most efficient strategies [180]. *Botrytis* spp. (necrotrophic pathogens) kills host cells via the secretion of toxins before taking host nutrients. In contrast, biotrophic pathogens (e.g., *Fusarium* spp. or *Phoma* spp.) do not kill the host cells but rather alter the host's metabolic and secretory systems to take nutrients from the host cells [181]. In related studies, the monolignol pathway is stimulated by fungal mycelial extracts in cell cultures of *Linum usitatissimum* [182]. In soybean and potato plants, microbial resistance can be induced by cultures of *Phytophthora*. Resistance to *Phytophthora* has been induced in *Capsicum annuum* using extracts obtained from microbial-rich composts [183]. The production of catharanthine, serpentine, and indole alkaloids (e.g., ajmalicine) in cell suspensions of *C. roseus* is induced by fungal cell-wall fragments [184]. These fragments also induce the production of raucaffrincine and 12-oxo-phytodienoic acid in *Rauwolfia canescens* [185]. Moreover, when fungal mycelia are used as elicitors in *Dioscorea deltoidea*, diosgenin content increases by up to 72% in the plant's cells [186]. Similarly, a more than eight-fold increase in the production of sanguinarine, morphine, and codeine has been observed in *P. somniferum* when fungal spores are used for the elicitation [187]. Furthermore, in cultures of *R. graveolens*, the production of antimicrobial alkaloids (e.g., acridone epoxide) increases by up to 100-fold when fungal polysaccharides are added [188].

### 5.1.4. Bacterial Elicitors

When bacterial elicitors are used in the hairy root culture of *Scopolia parviflora*, they activate the synthesis of scopolamine by inhibiting H6H (hyoscyamine 6β–hydoxylase) expression [46]. In addition, increased production of glycyrrhizic acid has been observed in the roots of *Taverniera cuneifolia* following treatment with *Rhizobium leguminosarum*, while significantly increased amounts of glycyrrhizic acid have been noted when *Bacillus cereus*, *Agrobacterium rhizogenes*, and *Bacillus aminovorans* are instead used for the elicitation [189]. In another study, Rhizobacterium induced the production of pseudohypericin and hypericin in the seedlings of *H. perforatum* [190].

### *5.2. Hormonal Elicitors*

In elicitation experiments, various plant hormones have also been tested [191]. Jasmonic acid and salicylic acid and their derivatives have been well studied because of the key roles they play in the response to plant defense (Table S2).

### 5.2.1. Jasmonates

The jasmonates, which comprise methyl jasmonate and jasmonic acid, are a group of compounds containing cyclopentanone that are key factors in the plant defense system. They have been shown to improve the production of SMs in in vitro cultures. For several plant secondary metabolic pathways, jasmonates constitute an essential class of elicitors; typically, they elicit SM biosynthesis when plants face unique environmental stresses [192]. Jasmonic acid is considered a signal molecule of plants following wounding or pathogen attack. In different cell cultures, jasmonic acid and its more active derivative methyl jasmonate can induce the development of a broad range of plant SMs such as rosmarinic acid, terpenoid indole alkaloid, and plumbagin [193,194]. For example, rosmarinic acid production in *Mentha piperita* can be elicited by jasmonic acid [193]. It also affects the production of anthocyanin in *V. vinifera* and production of plumbagin in the hairy roots of *Plumbago indica* [195]. Methyl jasmonate and jasmonic acid are also used as inducers of stilbene production in *V. vinifera* foliar cultures [196], *V. vinifera* cell cuttings [197], and *Vitis rotundifolia* root hair cutting cultures [198]. Furthermore, the addition of methyl jasmonate to *V. vinifera* in cell cultures encourages the production of anthocyanin compounds [199]. The use methyl jasmonate with transgenic technology also vastly increases the production

of tanshinone compounds in *Salvia miltiorrhiza* plant hairy roots [200], as well promoting withanone, and withanolide-A in hair root cultures of *Withania somnifera* [201]. The production of bacoside A, which is a notable triterpenoid saponin compound that shows nootropic healing activity in in vitro stem cultures of *Bacopa monnieri*, can also be increased by methyl jasmonate [202]. Furthermore, methyl jasmonate increases the production rate of andrographolide compounds in cell cultures of *Andrographis paniculate* [203], and increases production of soyasaponin in *Glycyrrhiza glabra* [204]. Finally, methyl jasmonate has been shown to increase the manufacturing output of paclitaxel in *Taxus cuspidate* and *Taxus canadensis* [205], and to enhance production of the raspberry ketone benzalacetone in the seedlings of *Rubus idaeus* [206].

### 5.2.2. Salicylic Acid

The formation of SMs in plants may also be induced by salicylic acid, which is well known for the systemic acquired resistance to several pathogens that it induces in plants. For example, salicylic acid can significantly enhance the development of tanshinone in *S. miltiorrhiza* hairy roots via transgenic technology [200]. Furthermore, in salicylic acid-provoked hairy root cultures of *W. somnifera*, the production of compounds with anolide A, withaferin A, and withanone was observed to have increased by [201]. In cell suspension cultures of *V. vinifera*, salicylic acid was able to enhance the production of stilbene [207]. It also triggers the production of alkaloids including vinblastine and vincristine in periwinkle plants [208], the tropane alkaloid scopolamine in *Brugmansia candida* plant hairy root cultures, and pilocarpine compounds in jaborandi plant leaves [209]. After treatment with salicylic acid, *Rubia cordifolia* shows enhanced production of anthraquinone [210]. Terpenoid base secondary metabolism can also be affected by salicylic acid treatment, and it initiates the accumulation of triterpenoid ginsenoside compounds in ginseng plants as well as production of glycyrrhizin in licorice plants [211,212]. Finally, monoterpene production is known to be enhanced by salicylic acid given suitable conditions [213].

### 5.2.3. Gibberellic Acid

The phytohormone gibberellin is also an active enhancer of SM production [214]. The diterpenoid compound gibberellic acid is synthesized in the same pathway used for artemisinin biosynthesis [215,216]. Various studies have investigated the use of gibberellic acid as an elicitor. For example, it was used to enhance the production of caffeic acid derivatives and tanshinones in hairy root cultures of *Echinacea pupurea* and *S. miltiorrhiza* [217,218]. Gibberellic acid has also been used in *Artemisia annua* to boost amounts of artemisinin [219]: it was used as an elicitor to enhance the biosynthesis of polyphenolics (flavonoids and phenolics) in suspended *Artemisia* cells.

## 6. Transcription Factors and Genes Associated with SMs

The inducible synthesis of SMs and the transcription of associated biosynthetic genes are strongly altered at various levels through transcriptional regulation via TFs. TFs are actually DNA binding proteins that attach to the promoter regions of target genes and alter the rate of transcriptional initiation via RNA polymerases. To regulate expression of enzyme genes, TFs can integrate internal and external signals and thereby monitor the accumulation of SMs. Regulation of genes associated with the SM biosynthesis pathway is under the influence of many TFs at various levels [220]. The identification of TFs and research on their regulatory mechanism in the SM biosynthesis pathway has increased in recent decades. In this section, we provide a review of the TFs associated with regulation of the SM pathway in various plant species (Figure 1).

### 6.1. WRKY TFs

The WRKY TFs family has been widely studied in plants under stress conditions. The inducible expression pattern of *WRKY* genes supports their participation in the modulation of defense-related SM biosynthesis. Members of the WRKY family have a 60-amino-acid

conserved domain that is responsible for gene regulation and overcoming the interaction with W-boxes of the target promoters. Plant WRKY TFs are largely involved in stress responses; they can be regulated by wound or jasmonic acid signaling, and they alter the expression of genes involved in the biosynthesis of various SMs such as alkaloids, terpenoids, and their subclasses [221]. For example, in tobacco plants, WRKY3 and WRKY6 are associated with the biosynthesis of volatile terpenes [222]. In cotton plants, WRKY1 regulates gossypol biosynthesis by binding to the promoter region of the gene involved in cadinene synthesis. In *Artemisia annua*, WRKY1 regulates artemisinin biosynthesis by binding to the promoter of the sesquiterpene synthesis gene [223]. WRKY1 was also studied in potato plants infected with late blight, in which it attached to the promoter site of the gene involved in hydroxycinnamic acid amide (HCAA) biosynthesis via modulation of the phenylpropanoid pathway [224]. The genes associated with HCAA, namely *ACT*, *DGK*, and *GL1*, are activated by the TF TaWRKY70 when a fungal biomass is encountered on wheat [225]. Similarly, HvWRKY23 promotes the expression of the genes *PAL*, *CHS*, and *HCT*, which are responsible for the induction of HCAA biosynthesis during Fusarium head blight infection [226]. In *V. vinifera*, VviWRKY24 enhances the expression of the *VviSTS29* gene, which is responsible for biosynthesis of resveratrol, which in turn confers antimicrobial resistance [227]. Furthermore, SsWRKY18, SsWRKY40, and SsMYC2 are responsible for regulation of abietane-type diterpene accumulation in *Salvia sclarea*; these compounds have antibacterial and antifungal properties [228]. It has been reported that silencing WsWRKY1 in *W. somnifera* significantly decreases the accumulation of phytosterol, which in turn decreases tolerance to bacteria, fungi, and insects [229]. In potato plants, StWRKY8 enhances the expression of the genes *TyDC*, *NCS*, and *COR2*, which are involved in the biosynthetic pathway responsible for benzylisoquinoline alkaloid production (these are antimicrobial agents) [230]. In addition, ZmWRKY79 increases the accumulation of phytoalexins in maize under stress conditions [231]. Furthermore, the expression of *dbat* in *Taxus chinenesis*, which is promoted by the TF TcWRKY1, results in increased accumulation of taxol [232]. In *C. roseus*, CrWRKY1 is induced by jasmonic acid to play a significant role in the biosynthesis of terpenoid indole alkaloids. CrWRKY1 also promotes the *TDC*, which is a gene involved in the biosynthesis pathway of terpenoid indole alkaloids [233]. Finally, a WRKY TF in *Panax quinquefolius*, namely PqWRKY1, enhances accumulation of triterpene ginsenoside (41). Thus, a single TF is not necessarily responsible for the concerted transcriptional activation of the whole biosynthesis pathway.

### 6.2. MYB TFs

Among the various TFs, MYBs are involved in the biosynthesis of SMs and participate in various biological processes in plants such as growth, reproduction, and stress responses. MYB TFs are characterized by different numbers of DNA-binding domains consisting of 50–53 amino acids of four imperfect repeats. They can be categorized into four sub classes, i.e., R1, R2, R3, and R4, depending on DNA-binding domain repeats. The R2R3 family of MYB TFs is significantly associated with the regulation of various SM pathways in different plant species. For instance, AtMYB113, AtMYB114, AtMYB75, and AtMYB90 in *Arabidopsis thaliana* are potentially involved in the regulation of anthocyanin residues via alteration of the phenylpropanoid pathway [234]. Additionally, MYB TFs may also be involved in the biosynthesis of GLs, flavonoids, HCAAs, and proanthocyanins. Moreover, in *Arabidopsis*, MYB29 and AtMYB76 are associated with the accumulation of aliphatic glucosinolate in aerial parts, whereas AtMYB34, AtMYB51, and AtMYB122 affect indole glucosinolate accumulation by altering the expression of tryptophan biosynthesis genes such as *CYP79B2*, *CYP79B3*, and *CYP83B1* [235]. Another study reported that the triple mutant of MYB-34, MYB-51, and MYB122 in *Arabidopsis* shows a considerable reduction in indole glucosinolate accumulation, which in turn increases susceptibility to *Plectosphaerella cucumerina* [236]. CsMYBF1 in citrus plants activates the expression of *CHS*, which is a gene involved in the biosynthesis of certain flavonoids [237]. Similarly, SbMYB8 from *Scutellaria baicalensis* plants enhances the expression of *CHS* in transgenic tobacco, which in

turn increases the regulation of flavonoids and enhances tolerance to drought stress [238]. A recent study showed that CsMYB2/CsMYB26 from *Camellia sinensis* plants binds to the promoter region of the genes *CsF3′H* and *CsLAR* and thereby enhances the accumulation of flavonoids, which are responsible for tolerance against blister blight disease caused by *Exobasidium vexans* [239]. In tobacco plants, MYB-JS1 is involved in the regulation of the phenylpropanoid pathway, whereas *MYB6* and *MYB12* in Asiatic hybrid lily plants are involved in the biosynthesis of anthocyanin [240]. Researchers have also reported that AtMYB11, AtMYB12, and AtMYB111 in *Arabidopsis* are significantly involved in the regulation of flavonoid biosynthesis, while heterologous expression of AtMYB TFs in various plants is also responsible for flavonoid accumulation [241]. Aside from the positive roles of AtMYBs in plant process, AtMYB75 negative affects flavonoid synthesis and significantly reduces the accumulation of kaempferol-3,7-dirhamnoside, which increases the susceptibility of plants to insect attack [242]. In contrast, MYB30, MYB55, and MYB110 in rice plants activate phenylpropanoid pathway genes, including *HCT*, *4CL3*, *C4H*, and *PAL*, which are responsible for the positive accumulation of HCCAs [243]. In poplar trees, PtMYB115 binds to the promoter region of the genes *ANR1* and *LAR3* to enhance their expression; this results in higher accumulation of proanthocyanin and increased resistance against *Dothiorella gregaria* [244].

### *6.3. bHLH TFs*

The bHLH TFs are potential regulators of stress response mechanisms and usually interact with MYB proteins to generate complexes that enhance the expression of specific genes. These TFs are 60-amino-acid proteins with bipartite conserved domains; the key residue at the N-terminal allows the binding of bHLH to DNA, while two alpha helices facilitate the association of helix-loop-helix with proteins to form a homo/heterodimeric complex. As important modulators of stress responses, the bHLH TFs regulate the biosynthesis of SMs such as anthocyanin, alkaloids, glucosinolate, diterpenoid phytoalexin, and saponins. Specifically, bHLH04, bHLH05, and bHLH06 interact with MYB51 to significantly regulate the biosynthesis of GLs in *Arabidopsis* plants [245]. In the biosynthesis of anthocyanin and flavonoids, bHLHs are widely involved in the regulation of the phenylpropanoid pathway. For instance, the bHLH proteins (e.g., GL3, eGL3, and TT8) bind with MYB in the presence of TTG1, thereby generating a complex of transcriptional regulation that is responsible for the regulation of anthocyanin biosynthesis genes [235]. In addition, MYC2, which is a key regulator of jasmonic acid signaling, belongs to the bHLH protein family and is involved in the regulation of SMs directly or indirectly. Ref. [246] showed that CrMYC2 is also a bHLH family protein; it attaches to the jasmonic acid-responsive element in the ORCA-3 promoter and enhances gene expression, which resultants in altered expression of the genes associated with alkaloid biosynthesis.

In apple plants, MdMYC2 can induce gene expression following jasmonic acid application; consequently, anthocyanin production is enhanced in a transgenic line. MdMYC2 is also known to persistently upregulate anthocyanin-related genes such as *DFR*, *F3H*, *UF3GT*, and *CHS*. It integrates gibberellic acid and jasmonic acid signaling pathways via an interaction with DELLA protein, leading to upregulation of sesquiterpene genes in *Arabidopsis* flowers. Occasionally, SMs are regulated by a complex of different TF family proteins such as the MYB–bHLH–WDR complex, which enhances anthocyanin and proanthocyanin regulation in *Arabidopsis* plants [247]. In rice plants, DPF has been identified as a bHLH TF that regulates the accumulation of diterpenoid phytoalexin via activation of diterpenoid phytoalexin-related genes [248]. In another study, NbbHLH1, NbbHLH2, and NbbHLH3 were associated with nicotine biosynthesis (according to expressed sequence tag screening) via virus-induced gene silencing; of these TFs, NbbHLH1 and NbbHLH2 produce positive regulation by binding to G-box elements in the putrescine N-methyltransferase promoter, whereas NbbHLH3 is a negative regulator [249]. Finally, TSAR1 and TSAR2 are bHLH TFs responsible for the regulation of saponin via activation of the gene *HMGR* in *Medicago truncatula* [250]. In short, bHLH TFs act independently or via interaction with other protein

families to enhance the biosynthesis of various biotic or abiotic stress-inhibiting SMs in various plants.

### 6.4. bZIP TFs

The bZIP TFs are dimeric, transcriptional enhancer proteins, with a conserved leucine zipper and positively charged DNA binding site. They typically function in the modulation of plant biological processes. Members of the bZIP TF family are associated with stress responses, largely with tolerance to ROS and osmotic imbalance. Various Yap-like bZIPs (Yap being a subset of the yeast bZIP protein), such as NapA, AtfA, Aoyap1, Afyap1, and Apyap1, have been found to function in *Aspergillus* species in response to osmotic, oxidative, nutrient, and drug stresses. Studies have shown that bZIP and oxidative stress regulate biosynthesis of SMs in fungi [251]. RsmA is a bZIP-like protein that suppresses the accumulation of SMs in a mutant line; in contrast, in an overexpressor line, RsamA is an active participant in restoration of SMs [252]. Specific bZIP proteins, including SmbZIP20, SmbZIP7, and AabZIP1, are also known to regulate pharmaceutically important SMs, namely, tanshinone and artemisinin, in *Salvia miltiorrhiza* and *Artemisia annua*, respectively [253]. In addition, the light responsive bZIP protein, HY5, is involved in the synthesis of anthocyanin under stress conditions, e.g., MdHY5 in apple plants either interacts with MdMYB10 or acts independently as an anthocyanin regulator [254]. Similarly, in tomato plants, when S1HY5 is attached to the G-box and ACGT region in the promoter site of *CHS* and *DFR*, it alters anthocyanin accumulation. It also modulates the expression of QH6 by binding to its binding site and enhancing the biosynthesis of monoterpenes [255]. Terpenoid phytoalexins, which are blast pathogen-resistant elements in rice, are also associated with bZIP proteins. For example, OsTGAP1 is a bZIP protein in rice that binds to the promoter site of the genes *OsKSL4* and *OsCPS4* and enhances biosynthesis of terpenoid phytoalexins. Additionally, OsTGAP1 moderates the expression of other genes related to terpenoid biosynthesis by modulating the MEP pathway [256]. In contrast to OsTGAP1, OsbZIP79 negatively regulates terpenoid phytoalexins by downregulating MEP pathway genes. Finally, the bZIP-like proteins AtfB and Apyap1 have been shown to enhance the regulation of aflatoxin in *Aspergillus parasiticus* [257].

### 6.5. AP2/ERF TFs

AP2/ERF TFs contain a ~60-amino-acid AP2 DNA binding domain; this conserved domain was first described in *A. thaliana* in the floral homeotic gene *APETALA2* (*AP2*). The AP2/ERF protein can be categorized into four subfamilies on the basis of variation in the additional conserved domain: AP2, ERF, RAV, and DREB (AP2 contains two AP2 domain, ERF contains one AP2 and a B-subfamily, RAV contains one AP2 and a B3 domain, and DREB contains one AP2 and an A-subfamily) [258]. AP2/ERF TFs are broadly involved in mediating plant stress responses via regulation of plant SMs [259]. ORCA and ORCA2 are AP2/ERF proteins that bind to the promoter sites of genes responsible for the biosynthesis of terpenoid indole alkaloids. Similarly, ORCA3 is a jasmonic acid-inducible AP2/ERF protein that attaches to the JERE element in the promoter site of two genes responsible for the biosynthesis of strictosidine synthase and tryptophan decarboxylase. Furthermore, AaERF1 and AaERF2 are two jasmonic acid-responsive AP2/ERF TFs found in *A. annus* that attached to the *CBF2* and *RAA* binding site of genes encoding CYP71AV1 and ADS; these TFs activate both the genes and thereby enhance the biosynthesis of artemisinin and artemisinic acids [260]. Jasmonic acid-responsive AP2/ERF TFs have also been isolated from tobacco, which synthesizes several nicotine alkaloids. For example, NtORC1/ERF221 and NtJAP1/ERF10 are AP2/ERF TFs found in tobacco that are positively involved in the transcriptional regulation of the gene *PMT*, which is associated with the nicotine biosynthesis pathway; overexpression of this genes significantly enhances nicotine and pyridine alkaloids [261]. The jasmonic acid-responsive elicitation of nicotine biosynthesis in tobacco also involves the action of AP2/ERF and bHLH proteins, suggesting that combined interaction of TFs is also capable of contributing to SM synthesis [262]. Furthermore,

the AP2/ERF protein has been shown to bind with CrPRX1 and play a crucial role in vinblastine biosynthesis. However, a lack of additional research has led to predictions that currently unidentified AP2/ERF TFs may be responsible for the regulation of bisindole alkaloids. GAME9, an important TF that also belongs to the AP2/ERF family, is responsible for increasing the biosynthesis of steroidal glycoalkaloids that offer protection against pathogens and insects. Specifically, GAME9 attaches to the binding site of steroidal glycoalkaloid-related genes, e.g., *HMGR*, *C5-SD*, *CAS*, *SGTs*, and *GAMEs*, and positively regulates these genes [263]. Some AP2/ERF TFs are also involved in the biosynthesis of phytoprotectants such as saponins. For instance, PnERF1 is a member of the Ap2/ERF family that binds to the promoter site of *HMGR*, *EPS*, *DS*, and *SS* genes to enhance the biosynthesis of saponins [264]. In addition, the Ap2/ERF GBERF1 regulates the expression of *PAL*, *C4H*, *C3H*, *CCR*, *HCT*, *CoMT*, and *F5H* genes to enhance the biosynthesis of lignin and protect against *Verticilium dahliae* [265]. Overall, the AP2/ERF TF family has been studied in detail because of its importance role in mitigating the effects of plant stress.

*6.6. NAC TFs*

NAC TFs constitute a large group of proteins that control various processes in plants including biotic and abiotic stress responses [266]. This family contains more than 100 members in rice and *Arabidopsis*. NAC proteins are characterized by a NAC domain (with a highly conserved N-terminus) that functions as a binding site; however, the C-terminus is a diverse domain that functions as a transcriptional regulator [266]. Members of this family are involved in the biosynthesis of SMs in response to stress. For instance, the NAC family member ANAC042 binds to the promoter site of camalexin biosynthetic genes, including *CYP71A12*, *CYP71A13*, and *CYP71B15*, to regulate camalexin and increase plant tolerance to *Alternaria brassicicola* infection [267]. Similarly, PtrNAC72 regulates the expression of the arginine decarboxylase gene (*ADC*), which is responsible for the biosynthesis of putrescine and can mediate ROS homeostasis in *Poncirus trifoliate* [268]. Furthermore, MfNAC from *Medicago falcate* has been reported to regulate glutathione (an antioxidant that can mediate ROS homeostasis) biosynthesis by controlling the expression of the glyoxalase-1 gene (*GLO1*) [269]. The NAC TF member HbNAC1 is found in *Hevea brasiliensis*; it binds to the *CACG* motif in the promoter region of small rubber particle protein, which is involved in the biosynthesis of latex [270]. Furthermore, PaNAC03 is a NAC TF that negatively regulates some genes in the flavonoid pathway, such as *CHS*, *F3′H*, and *LAR3*, and thereby enhances plant tolerance against *Heterobasidion annosum*. Similarly, ANACO3 downregulates *DFR*, *ANS*, and *LODX* genes, which are responsible for the biosynthesis of anthocyanin [271].

## 7. Conclusions

Being sessile in nature, plants develop various indigenous defensive mechanisms to cope with certain environmental stresses. SMs are natural tools used by plants to combat biotic and abiotic stresses. This review encloses an overview of the role of secondary metabolites in response to biotic and abiotic responses via regulation of transcriptional factors and genes involved in environmental stress tolerance. Typically, SMs are produced at low levels in plants; however, their biosynthesis induces with environmental stimuli to increase tolerance against stress condition. This review indicates that, some special SMs, genes and TFs regulates in specific biotic and a biotic stress. Many SMs directly inhibit pest and pathogen infection, some are herbivore deterrents, while most participate in maintaining redox balance by ROS scavenging (which also confers stress tolerance in plants). Some of the major TFs discussed in this review are closely associated with the expression of individual or multiple genes related to the SM biosynthesis pathway. In some cases, a network of two or TFs is required to regulate the associated genes. This review can provide a platform for additional experiments to explore some marker metabolites, genes, and TFs for the development of biotic and abiotic stress tolerant plants. Future research should focus on enhancing bioactive accumulation of SMs in stress conditions.

Future studies should make use of advanced techniques to test the effects of exogenous application of SMs on the tolerance of plants to biotic and abiotic stress.

**Supplementary Materials:** The following are available online at https://www.mdpi.com/article/10.3390/agronomy11050968/s1, Figure S1. Common structures for each classification of phenolic compounds and terpene. Table S1. Different types of elicitors used to enhance plant secondary metabolites production. Table S2. Hormonal elicitors used to produce specific plant secondary metabolites in plants.

**Author Contributions:** R.J. and S.A. wrote the manuscript, K.-M.K. conceptualized the study, and M.N. and L.L. reviewed and edited the manuscript. All authors have read and agreed to the published version of the manuscript.

**Funding:** This work was supported by the National Research Foundation of Korea Grant funded by the Korean Government (NRF-2021M3E5E6022715).

**Conflicts of Interest:** The authors declare that the research was conducted in the absence of any commercial or financial relationships that could be construed as a potential conflict of interest.

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
