# Peer review of "Plant Secondary Metabolite Biosynthesis and Transcriptional Regulation in Response to Biotic and Abiotic Stress Conditions"

_agronomy, doi:10.3390/agronomy11050968_

Round 1

Reviewer 1 Report

The Manuscript entitled “Plant secondary metabolite accumulation, transcriptional regulation, and production in response to biotic and abiotic stress” by Jan et al., comprehensively summarized the classification, function, biosynthetic pathway and regulation of plant SMs.

The manuscript is well-written, and I only have a few minor comments.

  • The authors should reconsider the title to highlight plant SM biosynthesis and transcriptional regulation.
  • References need to be added when describe the findings e.g. Lines 222, 232 and 260.
  • The name of TFs needs to be consistent, and the dash should be removed; e.g., WRKY-3 changes to WRKY3.
  • The Conclusion section can be improved, such as proposing the outstanding questions.

Author Response

  • The authors should reconsider the title to highlight plant SM biosynthesis and transcriptional regulation.
  • Reply: Thank you for your suggestion. We have revised the title accordingly.
  • References need to be added when describe the findings e.g. Lines 222, 232 and 260.

Reply: Citation and reference added in line 222, 232, and 260.

  • The name of TFs needs to be consistent, and the dash should be removed; e.g., WRKY-3 changes to WRKY3.

Reply: The name of all TFs was rearranged accordingly. Thank you.

  • The Conclusion section can be improved, such as proposing the outstanding questions.
  • Reply: Thank you for your kind suggestion, Conclusion section improved accordingly.

Reviewer 2 Report

The review manuscript is a well written and constructed, please find my comments and suggestion to improve its quality:

L69-81: please add reference

L71: revise "de" to "be" please

L158: In my opinion Furanocoumarins should be merged with the section "coumarins", since they belong to coumarins, any reason to separate them?

- I would recommend to add a general structure for each classifications, or provide a single Figure which demonstrate the chemical structures of the compound#s classification mentioned in the text

L223: please bring some examples for the sesquiterpenes

Table 1.: the reference numberings are not following of the text

- as you might know "Artemisinin" is the successful phytochemical with very potent antimalarial activity, please mention

L834: please re-write: " they are accumulated in response to stress to enhance plant tolerance to such adverse conditions."

References: please check all once more, e.g. in some cases title of the papers are bold, which is not based on the journal's instruction

Good luck!

Author Response

L69-81: please add reference

Reply: Reference added in line 81.

L71: revise "de" to "be" please

Reply: de changed to be thank you, line 71.

L158: In my opinion Furanocoumarins should be merged with the section "coumarins", since they belong to coumarins, any reason to separate them?

Reply: Thank you for your suggestion, Furanocoumarins and coumarins are structurally different from each other. Combination of furan and coumarins rings produces Furanocoumarins.  

- I would recommend to add a general structure for each classifications, or provide a single Figure which demonstrate the chemical structures of the compound#s classification mentioned in the text

Reply: Thank you for your kind suggestion; we included a supplementary figure for general structure of each group of phenolic and terpenes.

L223: please bring some examples for the Sesquiterpenes

Reply: Examples for Sesquiterpenes are included in the line 230, Thank you.

Table 1.: the reference numberings are not following of the text

Reply: We have arranged the reference accordingly.

- as you might know "Artemisinin" is the successful phytochemical with very potent antimalarial activity, please mention

Reply: Thank you for suggestion, antimalarial activity for artemisinin included in table.

L834: please re-write: “they are accumulated in response to stress to enhance plant tolerance to such adverse conditions."

Reply: thank you for your suggestion, the suggested sentence re-arranged now line # 835.

References: please check all once more, e.g. in some cases title of the papers are bold, which is not based on the journal's instruction

Reply: We revised and updated references according to the journal format, thank you.

Reviewer 3 Report

Review Comments

This review reviewed the information of how individual environmental factors affect the accumulation of secondary metabolites in plants during both biotic and abiotic stress conditions. Moreover, the applications of abiotic and biotic elicitors in culture systems as well as their stimulating effects on the accumulation of secondary metabolites were discussed. The authors reviewed the shikimate pathway and the aromatic amino acids produced in this pathway, which are the precursors of a range of secondary metabolites including terpenoids, alkaloids, and sulfur- and nitrogen-containing compounds. The biosynthesis of important metabolites altered by several genes related to secondary metabolite biosynthesis pathways were also reviewed. The authors also discussed the genes responsible for secondary metabolite biosynthesis in various plant species during stress conditions are regulated by transcriptional factors such as WRKY, MYB, AP2/ERF, bZIP, bHLH, and NAC. Therefore, this review provides some significantly useful insights, however some improvements and revisions are still required as shown below;

- The introduction should be improved by highlighting more cited works related to the topic of the review. Furthermore, the objectives of this review should be mentioned clearly in the introduction section as well.

- Figure 1 provides important mechanistic insights, however it should be discussed in more details.

- “Lignification is an appropriate response to pathogen infection and wounding; it sometime causes physical toughness that results in the indigestibility of plant tissue to pathogens from herbivores and insects”;   please rephrase this sentence to be more understandable and clear to the readers.

- Section 3.1.3 should be extended to provide more information on the importance and mechanistic actions of furanocoumarins.

- The English and language grammar should be revised and corrected throughout the whole manuscript.

- In Section 3.2.2, please provide the reference/citation for the following information “They are known for their defensive action against herbivores and ability to repel the feeding of mammals and insects”.

Also, provide the reference/citation for the following information “Sesquiterpenes also have a regulatory role in the initiation and preservation of bud and seed dormancy and can act as transcriptional activators”.

-  I could not see any references that cite the information provided in Section 3.2.5. please provide the citations and extend this section to cover the importance of polyterpenes.

- In Section 5.1, please provide the references/citations for the information provided about biotic elicitors.

- In Section 5.2, please provide the references/citations for the information provided about hormonal elicitors.

- Many sections in the manuscripts lack the references and citations, so the authors have to revise the whole manuscript and provide the citations for each information taken from the literature.

- The conclusions section should be revised to summarize the topics discussed in this review and provide future perspectives too.

- References should be revised and updated as per the discussed literature and the journal guidelines

Author Response

- The introduction should be improved by highlighting more cited works related to the topic of the review. Furthermore, the objectives of this review should be mentioned clearly in the introduction section as well.

Reply: Thank you for your valuable suggestion, we added some more information into the introduction, however there is high chance of repetition if we include more information.

- Figure 1 provides important mechanistic insights, however it should be discussed in more details.

 Reply: Thank you for your valuable suggestion. We have provided more detail about figure 1 and cited figure 1 in every suitable place in our MS.

- “Lignification is an appropriate response to pathogen infection and wounding; it sometime causes physical toughness that results in the indigestibility of plant tissue to pathogens from herbivores and insects”;   please rephrase this sentence to be more understandable and clear to the readers.

Reply: Re-arranged the sentence, thank you.

- Section 3.1.3 should be extended to provide more information on the importance and mechanistic actions of furanocoumarins.

 Reply: Thank you for your kind suggestion; we included some more information about furanocoumarin in the text line# 169-173.

- The English and language grammar should be revised and corrected throughout the whole manuscript.

 Reply: English editing certificate is attached to the resubmitted files.

- In Section 3.2.2, please provide the reference/citation for the following information “They are known for their defensive action against herbivores and ability to repel the feeding of mammals and insects”.

Reply: Reference and citation provided in line 239, thank you.

Also, provide the reference/citation for the following information “Sesquiterpenes also have a regulatory role in the initiation and preservation of bud and seed dormancy and can act as transcriptional activators”.

 Reply: Reference and citation provided in line 241, thank you.

-  I could not see any references that cite the information provided in Section 3.2.5. please provide the citations and extend this section to cover the importance of polyterpenes.

Reply: Reference and citation provided line 269. Although, there is very limited information present about polyterpenes, but still we included some information in line 270.

- In Section 5.1, please provide the references/citations for the information provided about biotic elicitors.

 Reply: Reference and citation added in line 523, thank you.

- In Section 5.2, please provide the references/citations for the information provided about hormonal elicitors.

 Reply: Reference and citation added in line 573, thank you.

- Many sections in the manuscripts lack the references and citations, so the authors have to revise the whole manuscript and provide the citations for each information taken from the literature.

 Reply: Thank you for your kind suggestion, due to the huge number of references we removed some of the references related to a well-known facts, however according to your suggestion and other reviewers comments we included some reference such as in line, 71, 81, 222, 232, 260. 

- The conclusions section should be revised to summarize the topics discussed in this review and provide future perspectives too.

 Reply: Thank you for your valuable suggestion, conclusion section rearranged accordingly.

- References should be revised and updated as per the discussed literature and the journal guidelines

 Reply: We revised and updated references according to the journal format, thank you.

Round 2

Reviewer 3 Report

The authors have improved the manuscript as per my suggested comments